# SMAD2/3 signaling in the uterine epithelium controls endometrial cell homeostasis and regeneration

Maya L. Kriseman[1,2,3], Suni Tang[1,2], Zian Liao[1,2,4], Peixin Jiang[1,5], Sydney E. Parks[1,2,6], Dominique I. Cope[1,2], Fei Yuan [1,2], Fengju Chen[7,8], Ramya P. Masand[1], Patricia D. Castro [1], Michael M. Ittmann [1], Chad J. Creighton [7,8], Zhi Tan[1,2,9] & Diana Monsivais [1,2,6,8✉]

The regenerative potential of the endometrium is attributed to endometrial stem cells; however, the signaling pathways controlling its regenerative potential remain obscure. In this study, genetic mouse models and endometrial organoids are used to demonstrate that SMAD2/3 signaling controls endometrial regeneration and differentiation. Mice with conditional deletion of SMAD2/3 in the uterine epithelium using Lactoferrin-iCre develop endometrial hyperplasia at 12-weeks and metastatic uterine tumors by 9-months of age. Mechanistic studies in endometrial organoids determine that genetic or pharmacological inhibition of SMAD2/3 signaling disrupts organoid morphology, increases the glandular and secretory cell markers, FOXA2 and MUC1, and alters the genome-wide distribution of SMAD4. Transcriptomic profiling of the organoids reveals elevated pathways involved in stem cell regeneration and differentiation such as the bone morphogenetic protein (BMP) and retinoic acid signaling (RA) pathways. Therefore, TGFβ family signaling via SMAD2/3 controls signaling networks which are integral for endometrial cell regeneration and differentiation.

[1] Department of Pathology and Immunology, Baylor College of Medicine, Houston, TX 77030, USA. [2] Center for Drug Discovery, Baylor College of Medicine, Houston, TX 77030, USA. [3] Division of Reproductive Endocrinology and Infertility, Baylor College of Medicine, Houston, TX 77030, USA. [4] Department of Molecular and Human Genetics, Baylor College of Medicine, Houston, TX 77030, USA. [5] Department of Thoracic/Head and Neck Medical Oncology, the University of Texas MD Anderson Cancer Center, Houston, TX 77030, USA. [6] Cancer and Cell Biology Program, Baylor College of Medicine, Houston, TX 77030, USA. [7] Department of Medicine, Baylor College of Medicine, Houston, TX 77030, USA. [8] Dan L. Duncan Comprehensive Cancer Center, Baylor College of Medicine, Houston, TX 77030, USA. [9] Department of Pharmacology and Chemical Biology, Baylor College of Medicine, Houston, TX 77030, USA. ✉email: dmonsiva@bcm.edu

The endometrium is the mucosal inner lining of the uterus that is under the cyclical control of the steroid hormones, estrogen (E2) and progesterone (P4), and holds the potential to undergo hundreds of cycles of regeneration throughout a woman's reproductive lifespan. The regenerative potential of the endometrium is conferred by the presence of unique stem and progenitor cells that are located throughout the endometrium to aid in rapid endometrial regeneration after menstruation[1–4]. Pharmacological inhibition of the transforming growth factor β (TGFβ) signaling pathway is critical for maintaining the proliferation and regeneration of endometrial mesenchymal stem cells and endometrial epithelial organoids, indicating that TGFβ family signaling is a critical pathway in endometrial regeneration and repair[5–8]. However, the underlying mechanisms controlled by TGFβ signaling in the endometrium are not well-understood.

The TGFβ signaling pathway is comprised of a variety of secreted ligands, inhibitors, cell surface receptor kinases, and the SMAD2/SMAD3 and SMAD4 transcription factors that are activated via phosphorylation and translocate to the nucleus to control gene expression[9]. This signaling pathway controls key developmental processes, such as cell migration, differentiation, and proliferation, and is dysregulated in various cancer subtypes. Mutations in TGFβ pathway members have been identified in various tumor subtypes and correlate with metastasis-associated genes and decreased patient survival[10].

In the female reproductive tract, TGFβ is important for the integrity and development of the uterus and ovary and controls processes during early pregnancy and throughout gestation, such as endometrial receptivity, decidualization, and placentation[11–16]. Components of the TGFβ signaling pathway also act as tumor suppressors. Mouse models have shown that conditional deletion of the TGFβ receptor, ALK5, or the SMAD2 and SMAD3 transcription factors in the endometrial epithelium and stroma[17] results in aggressive metastatic endometrial cancer and death[18,19]. In addition, endometrial cancer mouse models with conditional inactivation of phosphatase and tensin homolog (Pten) and the AT-rich interaction domain 1 A (Arid1a) genes, demonstrate aberrant TGFβ signaling, further supporting that inactivation of the TGFβ pathway contributes to the metastatic potential of endometrial tumors.

In this study, we define the epithelial-specific contribution of SMAD2/3 to endometrial function by conditionally ablating the Smad2 and Smad3 transcription factors with Lactoferrin-iCre (Ltf-cre). Using epithelial organoid cultures from the endometrium, we uncover the signaling mechanisms downstream of TGFβ that control endometrial cell regeneration and are critical for endometrial regeneration and homeostasis.

## Results

### Identification of TGFβ signaling pathway mutations in endometrial cancer.
We profiled the data from uterine tumors deposited to the cBioPortal of Cancer Genomics for mutations of the TGFβ signaling pathway[20,21] and identified several mutations in TGFβ-related receptors (TGFBR1, TGFBR2, ACVR1B, ACVR1C, ACVR2A, ACVR2B) and transcription factors (SMAD2, SMAD3, SMAD4) (Supplementary Fig. 1a). We found that among 894 patient tumors, ~20% harbored genetic abnormalities in TGFβ-related genes. Of the 52 mutations located in SMAD2, 41 were missense and 11 were truncating mutations (Supplementary Fig. 1b). SMAD3 harbored 41 mutations, while SMAD4 had 36 mutations (Supplementary Fig. 1c, d).

Receptors in the TGFβ signaling pathway also displayed several mutations; TGFBR1 had 42 mutations, while TGFBR2 had ~28 mutations (Supplementary Fig. 1e, f). ACVR1B and ACVR1C

were also mutated in uterine tumors, showing ~34 and 31 mutations each (Supplementary Fig. 1g, h). ACVR2A had the highest mutation load, harboring 80 total mutations, including 32 frameshift mutations affecting a single amino acid (K437R) (Supplementary Fig. 1i). ACVR2A K437R is a frequently occurring mutation in cancers that likely perturbs activin/SMAD2/3 signaling due to alterations in the C-terminal domain[10,22–24]. We also found that ACVR2B had 30 mutations (Supplementary Fig. 1j). Hence, these analyses indicate that genetic alterations affecting the normal function of the TGFβ signaling pathway are present in uterine cancers and may contribute to the onset and progression of malignancy.

### Generation of a mouse model with conditional deletion of Smad2 and Smad3 in the uterine epithelium.
Given the TGFβ signaling pathway alterations found in human endometrial tumors, we generated mice with epithelial-specific inactivation of Smad2[25] and Smad3[15] using Ltf-cre[26] ("Smad2/3 cKO" mice) to investigate the in vivo roles of SMAD2 and SMAD3 signaling in the luminal uterine epithelium, (Fig. 1a). Ltf-cre activity has been demonstrated in the uterine epithelium of the adult mouse uterus beginning at ~60 days of age, although cre-activation can be achieved at younger ages with administration of estradiol (E2)[26,27]. To obtain tissue-specific deletion of Smad2 and Smad3, we treated mice with 100 ng of E2 at days 21 and 22 of age. We verified that the specific exons were deleted in the uterine epithelium of Smad2/3 cKO mice at 6 weeks of age (Fig. 1b, c) and confirmed that total SMAD2 and SMAD3 protein levels were deleted from the uterine epithelium of Smad2/3 cKO mice (Fig. 1d, Supplementary Fig. 2). Phosphorylated SMAD2/3 (pSMAD2/3) immunoreactivity was detected in the epithelium and stroma of the control uterine tissues (Fig. 1e, f, red arrows); and while the uterine stroma of Smad2/3 cKO displayed pSMAD2/3 immunoreactivity, none was detected in the epithelial compartment (Fig. 1g, h, yellow arrows). These results indicated that we successfully generated a mouse model with double conditional deletion of SMAD2 and SMAD3 in the uterine epithelium using Ltf-cre.

### Mice with double SMAD2 and SMAD3 conditional deletion develop endometrial hyperplasia and lose progesterone receptor expression.
To determine how loss of SMAD2 and SMAD3 contribute to the integrity of the uterine epithelium, we analyzed the uterine tissues of control and Smad2/3 cKO mice at 12 weeks and 6 months of age (Supplementary Fig. 3). To visualize the uterine compartments, we immunostained uterine cross-sections with the epithelial cell marker, E-cadherin (CDH1, labeled with a green fluorophore) and with the myometrial marker, smooth muscle actin (SMA, labeled with a red fluorophore). We found that unlike control mice, Smad2/3 cKO mice displayed more uterine epithelial folds (Supplementary Fig. 3a, b) and loss of uterine epithelial progesterone receptor (PR) expression (Supplementary Fig. 3c–f). Immunostaining with the glandular marker, FOXA2, showed that endometrial glands from both the control and Smad2/3 cKO mice expressed FOXA2 (Supplementary Fig. 3g–j). By 6 months of age, Smad2/3 cKO had developed endometrial hyperplasia, as evident by the presence of E-cadherin-positive epithelial folds (Supplementary Fig. 3k, l) with a decrease of PR within the uterine epithelium (Supplementary Fig. 3m–p). Similar to the 12-week timepoint, FOXA2 immunoreactivity was detected in both Smad2/3 cKO and control mice at 6 months of age (Supplementary Fig. 3q–t).

We also assessed how single SMAD2 or SMAD3 deletion with Ltf-cre affected endometrial architecture by analyzing the uterine morphology of Smad2 cKO and Smad3 cKO mice at 6 months

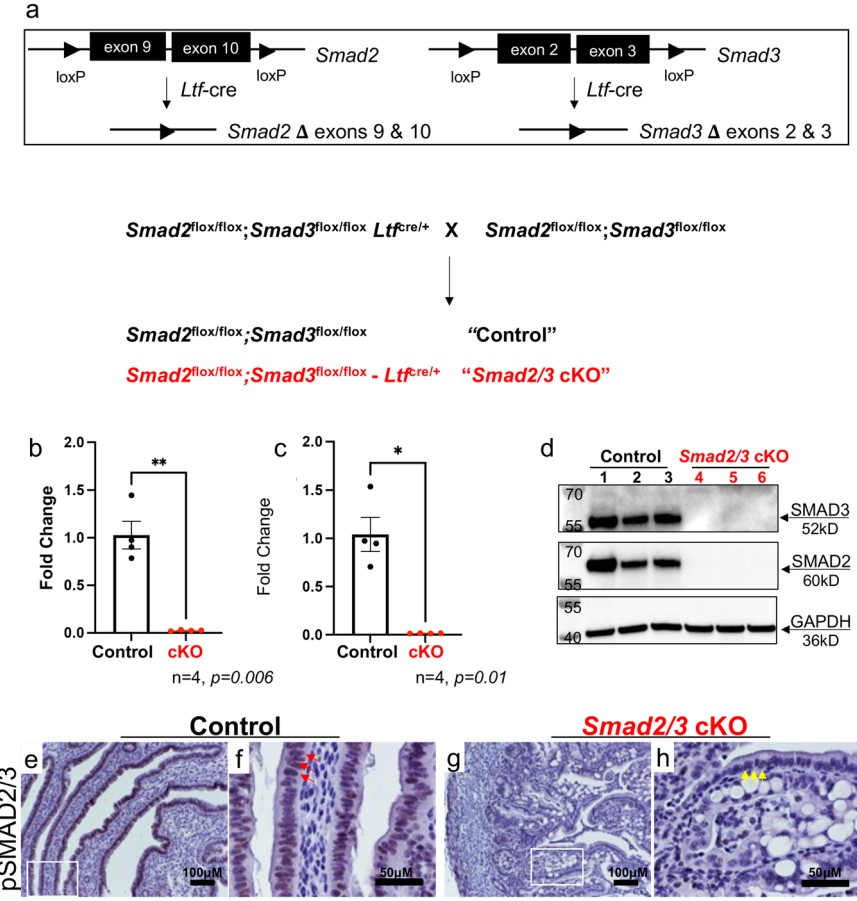

**Fig. 1 Generation of mice with double conditional deletion of SMAD2 and SMAD3 using *Ltf*-cre. a** Diagram showing the schematic used to obtain conditional deletion of SMAD2 and SMAD3 in the uterine epithelium using *Ltf*-cre. **b–h** Confirmation that effective deletion of the *Smad2* and *Smad3* floxed exons and protein levels were decreased in the uterine epithelium of *Smad2/3* cKO mice at the mRNA level (**b**, **c**, $n = 4$ per genotype) and in protein lysates from purified epithelium (**d**, $n = 3$ per genotype). Lanes in (**d**) were generated from the same blot, which was sequentially probed and stripped with each of the indicated antibodies. **e–h** Immunohistochemical analysis of phosphorylated SMAD2 and SMAD3 (pSMAD2/3) in uterine cross-sections of control ($n = 3$) and *Smad2/3* cKO mice ($n = 3$). Red arrows (**f**) show positively stained cells in uterine epithelial cells of controls, while yellow arrows (**h**) highlight the unstained epithelial cells in *Smad2/3* cKO uterus. Histograms represent mean ± SEM analyzed by an unpaired two-tailed *t*-test.

and 9 months of age (Supplementary Fig. 4). The uterine compartments of 6-month-old virgin mice were visualized with cytokeratin 8 (CK8, epithelial marker) and smooth muscle actin (SMA, myometrial marker), which showed that mice with single SMAD2 or SMAD3 deletion demonstrated no morphological differences when compared to the control mice (Supplementary Fig. 4a–f). Likewise, analysis of 9-month-old *Smad2* cKO and *Smad3* cKO uteri with E-cadherin staining (CDH1, epithelial marker) showed no morphological differences relative to control (Supplementary Fig. 4g–l). There was also no difference in FOXA2 or PR expression in the uterine tissues of controls and the single *Smad2* cKO and *Smad3* cKO mice (Supplementary Fig. 4m–r). Overall, we concluded that double conditional deletion of SMAD2 and SMAD3, and not single deletions of either gene, from the uterine epithelium of mice resulted in the development of endometrial hyperplasia and loss of epithelial, but not stromal, PR expression.

**Mice with double SMAD2 and SMAD3 conditional deletion develop metastatic uterine tumors.** We observed that unlike their age-matched control littermates, *Smad2/3* cKO female mice began to perish as early as 7 months of age. Upon gross examination, *Smad2/3* cKO mice developed bulky uterine tumors (Fig. 2a, b) with lung nodules (Fig. 2c, d). We observed that 77.8%

of the mice (14/18) had bulky uterine tumors and that 66.7% (12/18) harbored visible lung nodules. Histological examination of the uterine tumors showed endometrial infiltration into the underlying myometrium, suggestive of uterine carcinoma (Fig. 2e), while the lung nodules showed a border and distinct morphology from that of the normal lung (Fig. 2f).

While *Ltf*-cre shows the most potent recombinase activity within the uterine epithelium, a small number of cells with *Ltf*-cre activity have been identified in extra-uterine tissues, including the lung[26]. In our previous studies, conditional deletion of TGFBR1/ALK5 or SMAD2 and SMAD3 using progesterone receptor-cre resulted in aggressive uterine tumors with distant lung metastases[18,19]. To determine whether the lung nodules were metastases from the uterine tumors or independent lung lesions, we performed immunohistochemistry (IHC) on cross-sections of the lung nodules and uterine tumors using estrogen receptor alpha (ERα) as a uterine cell marker and transcription termination factor (TTF1) as a lung cell-specific marker (Fig. 2g–n). We found that the lung nodules had clearly demarcated ERα staining patterns with nuclear localization similar to that of the cells within the uterine tumors (Fig. 2g–j). Analysis of the lung-specific marker, TTF1, showed clear nuclear staining within the cells of the normal lung (Fig. 2k, l, yellow arrows). However, no TTF1-positive cells were detected within the lung nodules (Fig. 2k, l) or in the uterine tumors (Fig. 2m, n).

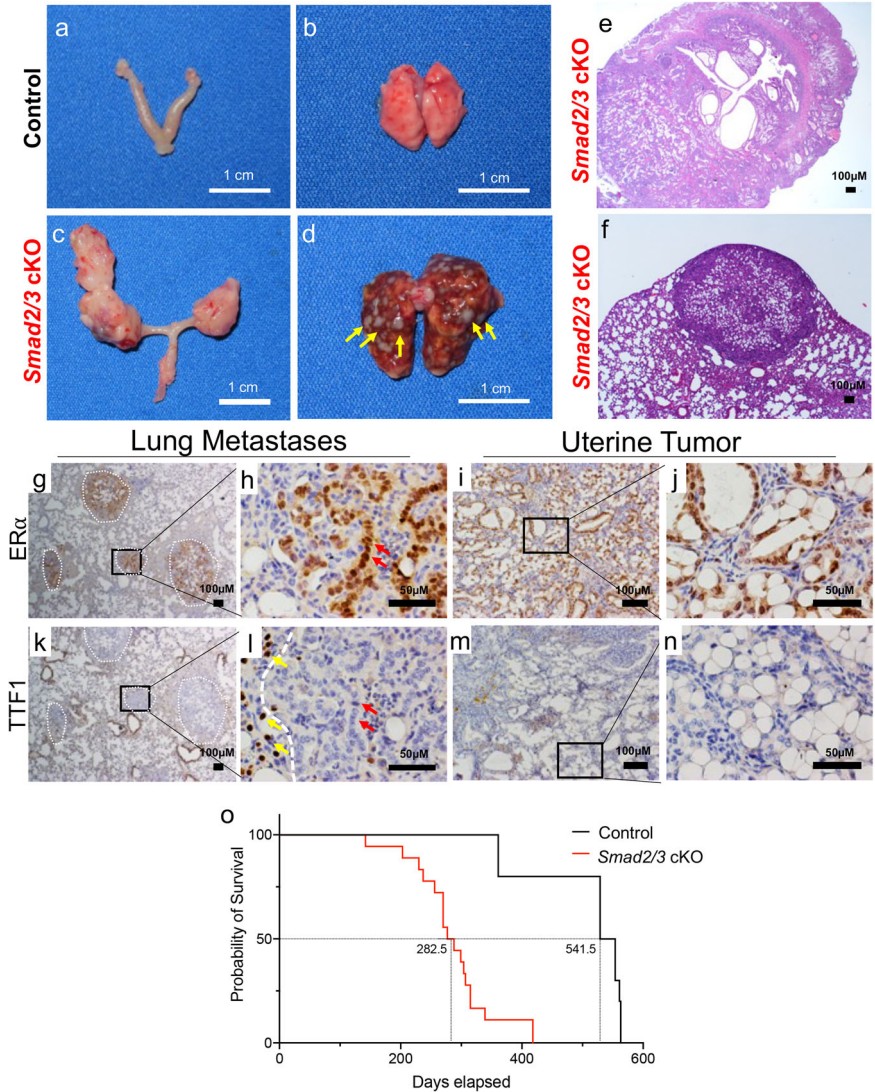

**Fig. 2 Conditional deletion of SMAD2/3 results in metastatic endometrial tumors and death. a, b** Gross uterus of control (**a**) and *Smad2/3* cKO (**b**) mice at 9 months of age. *Smad2/3* cKO mice showed the presence of uterine masses. **c, d** Lungs dissected from control (**c**) and *Smad2/3* cKO mice (**d**), showing that the lungs of the *Smad2/3* cKO mice developed metastatic nodules (yellow arrows). **e, f** Cross-sections of the uterine tumor (**e**) and metastatic nodules (**f**) from *Smad2/3* cKO mice stained with Hematoxylin and Eosin (H&E). **g–j** Immunohistochemistry of ERα in the lung nodules (**g, h**) or uterine tumors (**i, j**) from *Smad2/3* cKO mice. Expression of ERα is observed in the uterine tumors (**i, j**), and in the lung nodules (outlined by white dotted circles), but not in the adjacent normal tumor tissue. **k–n** Immunohistochemistry of the lung cell marker, TTF1, in lung (**k, l**) and uterine tumor cross-sections (**m, n**) showing that neither the uterine tumors (**m, n**) nor metastatic nodules (**k, l**) express TTF1. However, the normal lung cells adjacent to the lung nodules do express TTF1 (**l**, yellow arrows). Red arrows in (**h, l**) show ERα positive cells in the lung nodules (**h**) that are TTF-negative in a sequential section (**l**). **o** Survival analysis comparing the survival of control mice (50% survival, 541.5 days) to *Smad2/3* cKO mice (50% survival, 282.5 days).

In addition, when serial sections of the lung nodules were stained with TTF1 and ERα, cells that were ERα positive did not show TTF1-expression (Fig. 2h, l, red arrows). These results indicated that uterine-epithelial deletion of SMAD2 and SMAD3 with *Ltf*-cre resulted in uterine tumors with distant metastases to the lungs. These tumors and metastases led to the premature death of the mice, resulting in reduced survival for the *Smad2/3* cKO mice relative to the controls (Control: 541.5 ± 25.3 vs. *Smad2/3* cKO: 282.5 ± 15.7-day survival) (Fig. 2o). Hence, an intact SMAD2 and SMAD3 signaling program within the uterine epithelium is required for normal endometrial function and homeostasis.

**Uterine tumor development in SMAD2 and SMAD3 double conditional knockout mice is driven by estrogen.** In women, endometrial tumors are typically classified according to

histological and molecular criteria, including their response to estrogen (E2) and their ER and PR expression status[28]. To determine the effect of E2 on tumor development, we ovariectomized 8-week-old control and *Smad2/3* cKO mice and implanted a placebo or 90-day-release E2 pellet (Fig. 3). We found that *Smad2/3* cKO mice containing the E2 pellet perished ~34 days sooner than mice in the other groups (*Smad2/3* cKO + E2, *n* = 4: 120 ± 10.2 days vs. all other groups, *n* = 4 per group: 154 days). As expected, administration of E2 to ovariectomized control mice increased gross uterine weight (control + E2, *n* = 4, 0.108 ± 0.038 g) compared to that of controls and *Smad2/3* cKO mice without E2 (control + placebo, *n* = 4, 0.019 ± 0.003 g; *Smad2/3* cKO + placebo, *n* = 4, 0.025 ± 0.002 g). However, administration of E2 pellets to *Smad2/3* cKO drove tumorigenesis, resulting in increased uterine weight (*Smad2/3* cKO, *n* = 4, 2.99 ± 0.66 g), tumors in 100% of the mice (4/4), and

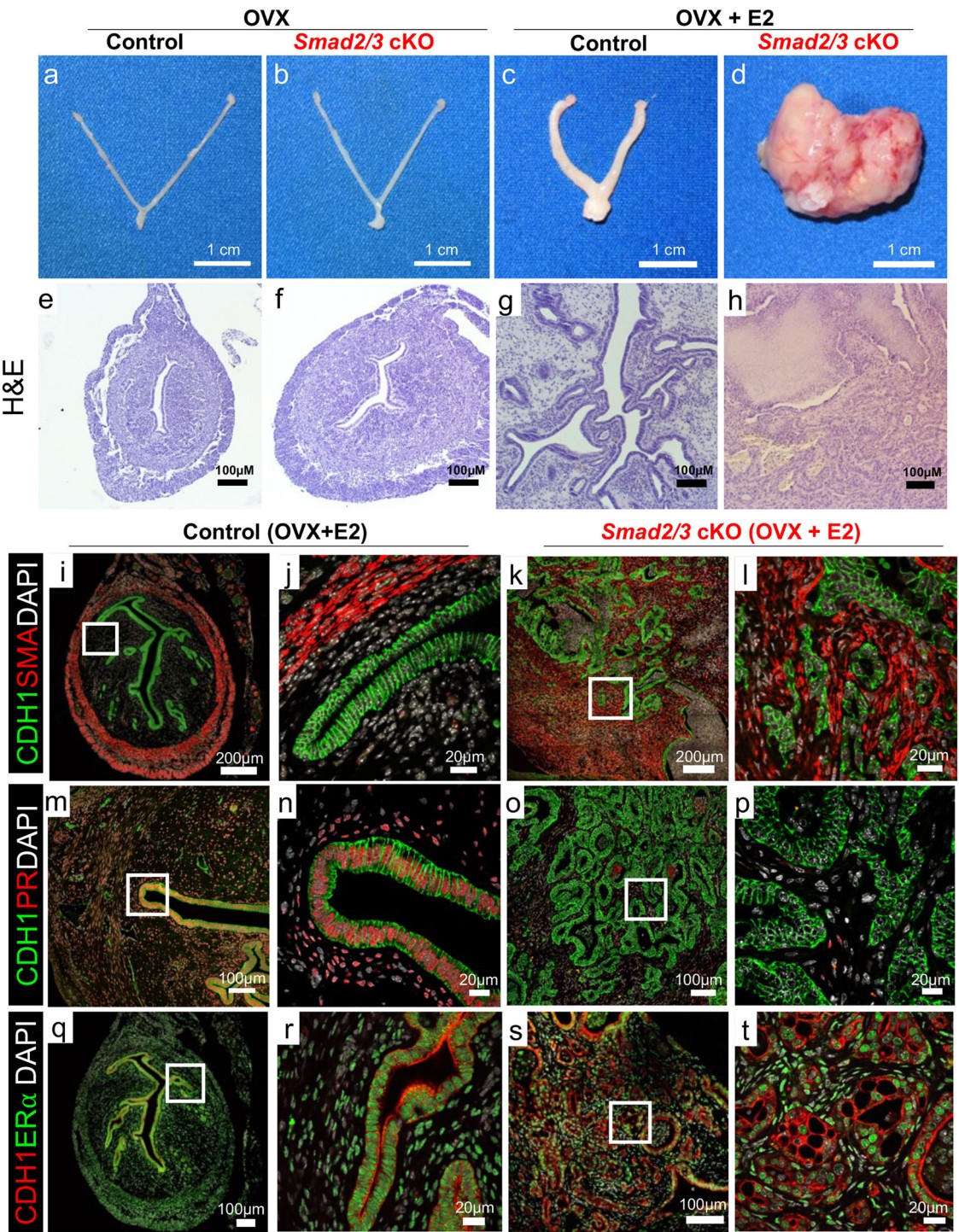

**Fig. 3 Endometrial tumor development is estrogen dependent in *Smad2/3* cKO mice. a–d** Gross uteri from adult control (**a**, **c**) and *Smad2/3* cKO mice (**b**, **d**) collected 3 months after ovariectomy (OVX) and treated without (**a**, **b**) or with estradiol releasing pellets (**c**, **d**). Only the *Smad2/3* cKO mice that received the estradiol treatment developed tumors. **e–h** H&E stained uterine cross-sections stained of control (**e**, **g**) and *Smad2/3* cKO (**f**, **h**) that were ovariectomized and treated without (**e**, **f**) or with estradiol (**g**, **h**). **i–t** Immunostaining of control (**i**, **j**, **m**, **n**, **q**, **r**) and *Smad2/3* cKO mice (**k**, **l**, **o**, **p**, **s**, **t**) cross sections following OVX + E2 treatment. **i–l** Tissue sections were stained with the epithelial cell marker, E-cadherin (CDH1, green) and smooth muscle actin (SMA, red). Compared to controls (**i**, **j**) sections from *Smad2/3* cKO mice (**k**, **l**) show disordered epithelial cell and smooth muscle layers. **m–p** Tissue sections were stained with E-cadherin (CDH1, green) and progesterone receptor (PR, red). PR can be seen in the nuclei of the control mice (**m**, **n**) but not in the epithelium of *Smad2/3* cKO mice (**o**, **p**). **q–t** Uterine cross sections were stained with E-cadherin (CDH1, red) and estrogen receptor α (ERα, green) antibodies. Cross sections from control (**q**, **r**) and *Smad2/3* cKO (**s**, **t**) mice were positive for ERα. Nuclei were stained with DAPI (white). H&E and immunostaining experiments were performed in samples from at least 3 control and 3 *Smad2/3* cKO mice.

lung metastases in 50% of the mice (2/4) (Fig. 3a–d, Supplementary Table 1). This result indicated that tumor development in *Smad2/3* cKO mice was E2-dependent.

Cross-sections of the uteri revealed that the untreated ovariectomized control, *Smad2/3* cKO mice, and the ovariectomized controls treated with E2, retained a properly organized endometrial compartment (Fig. 3e–g). However, ovariectomized *Smad2/3* cKO mice treated with E2 displayed expansion of the epithelial cells in the tumor mass (Fig. 3h). Immunostaining of the uterine cross-section showed that while smooth-muscle actin staining (SMA, red) was restricted to the outer myometrium and E-cadherin-positive cells (CDH1, green) were in organized epithelial structures, the uterine architecture of the *Smad2/3* cKO mice treated with E2 was disorganized (Fig. 3i–l). We also observed that while the epithelium and stroma of the control mice expressed PR, few stromal but no epithelial cells expressed PR in the *Smad2/3* cKO + E2 tumors (PR, red; Fig. 3m–p). Immunostaining with the estrogen receptor α antibody (ERα, green) showed that the stroma and epithelium of both control and *Smad2/3* cKO mice expressed ERα (Fig. 3q–t), suggesting that the tumor's estrogen response was occurring via the transcriptional activity of ERα.

Because *Ltf*-cre activity is induced by E2, we confirmed this finding in a separate cohort *Smad2*^flox/flox;*Smad3*^flox/flox mice administered with intrauterine adenoviral-cre (Ad-empty or Ad-cre) following ovariectomy and placebo or E2 pellet implantation (Supplementary Fig. 5a). This allowed us to assess the role of E2 in a mouse line with Ad-cre-induced SMAD2/3 deletion independent of *Ltf*-cre activation. We used a strategy of intrauterine Ad-cre injection that was previously shown to effectively target the major cell types within the uterus[29,30]. As expected, administration of E2 to mice injected with Ad-empty increased uterine weight (Ad-empty + placebo, $n = 2$, $0.06 \pm 0.023$ g vs. Ad-empty + E2, $n = 3$, $0.12 \pm 0.05$ g) (Supplementary Fig. 5b). Administration of Ad-cre to mice with a placebo pellet did not affect uterine weight (Ad-cre + placebo, $n = 3$, $0.02 \pm 0.006$), however Ad-cre in the presence of E2 caused uterine tumor development in 1/3 of mice (Ad-cre + E2, $n = 3$, $1.36 \pm 1.9$ g) (Supplementary Fig. 5b). Histological analyses confirmed that only the mice treated with Ad-cre + E2 developed uterine tumors with glandular infiltration into the underlying myometrium (Supplementary Fig. 5b). Therefore, these results confirmed that tumor development in *Smad2/3* cKO mice is dependent on E2 signaling.

**Genetic and pharmacological SMAD2/3 inactivation affects organoid morphology and differentiation.** To determine the signaling pathways that are abrogated in the epithelium of *Smad2/3* cKO mice, we established endometrial epithelial organoids with genetic or pharmacological inhibition of SMAD2/3 signaling. It was previously shown that long-term culture of endometrial epithelial organoids from mice require, in part, inhibition of SMAD2/3 signaling[7]. Although this was achieved by the addition of the type 1 TGFβ receptors (ALK4/5/7) A83-01[31], the downstream signaling pathways affected by the inhibitor remain unknown. To uncover these biological networks, we cultured endometrial organoids from *Smad2/3* cKO and control mice in the presence or absence of A83-01 to obtain genetic or pharmacological suppression of TGFβ signaling (Fig. 4a). After 3–5 passages (~3–5 weeks), endometrial organoids cultured with A83-01 or from *Smad2/3* cKO mice, developed an abnormal "dense" morphology compared to those organoids from control mice grown without A83-01, which retained a round "cystic" morphology (Fig. 4b, c). We quantified the development of cystic vs. dense organoids across the three conditions (Control + Vehicle, Control + A83-01, *Smad2/3* cKO)

over five passages (Fig. 4d–f). We found that the morphology of control organoids cultured without A83-01 retained a round cystic morphology over 5 passages, while ~27-43% of control organoids cultured with A83-01 began to develop a lobular dense morphology starting at approximately passage 3 (Fig. 4e). Endometrial organoids from *Smad2/3* cKO mice began to develop a dense morphology as early as passage 1 (~11%), with approximately ~44% of these organoids displaying a dense morphology by passage 4 (Fig. 4f).

Histological analysis of the endometrial organoids showed that while the organoids from control mice grown without A83-01 retained a single layer of epithelium (Fig. 4g, j), control organoids cultured with A83-01 and *Smad2/3* cKO organoids displayed a more complex organization with enlarged secretory-like cells (Fig. 4h, i, k, l). Immunostaining with cytokeratin 8 (CK8) and mucin 1 (MUC1) showed more prominent MUC1 expression in organoids from control mice with A83-01 and *Smad2/3* cKO mice than organoids from control mice cultured with vehicle (Fig. 4m–o). Likewise, organoids from controls cultured with A83-01 or from *Smad2/3* cKO mice displayed more prominent expression of the glandular cell marker, FOXA2, than those from control mice cultured with the vehicle (Fig. 4p–r). These results suggested that the genetic or pharmacological inhibition of TGFβ signaling increased differentiation of the organoids toward secretory-like cells. Development of mouse epithelial organoids with a "dense" morphology was previously observed to be WNT-dependent, suggesting that inhibition of TGFβ signaling inhibits a similar pathway[7].

**Genetic or pharmacological inhibition of TGFβ signaling elevates BMP and retinoic acid signaling in endometrial organoids.** To characterize the gene expression pathways controlled by the inhibition of TGFβ signaling, we performed RNA sequencing (RNAseq) of endometrial organoids from the three conditions described above (control + vehicle, control + A83-01, and *Smad2/3* cKO) (Fig. 5). Differential gene expression was calculated between the groups as follows, (1) control + A83-01 vs. control + vehicle, or (2) *Smad2/3* cKO vs. control + vehicle, using a cutoff of >1.4-fold change, <0.714-fold change, and 0.01 false discovery rate (FDR). Using these parameters, we identified that 569 genes were upregulated and 570 were downregulated in comparison (1) control + A83-01 vs. control + vehicle (Fig. 5a, b); while 912 genes were upregulated and 865 downregulated in comparison (2) *Smad2/3* cKO vs. control + vehicle organoids (Fig. 5a, c).

Gene ontology analysis of upregulated genes in the *Smad2/3* cKO organoids showed enrichment of networks involved in retinol metabolism (*adj. $p = 1.01*10^{-3}$*), such as lecithin retinol acyltransferase (*Lrat*), cytochrome P450 family subfamily A member 1 (*Cyp26a1*), and aldehyde dehydrogenase 1 family member A2 (*Aldh1a2*) (Fig. 5d and Supplementary Data 1). We also observed that BMP-activated genes, inhibitor of DNA binding 1 (*Id1*), inhibitor of DNA binding 3 (*Id3*) and inhibitor of DNA binding 4 (*Id4*) were upregulated in the organoids of *Smad2/3* cKO mice (Fig. 5c). This suggests that the decreased SMAD2/3 signaling in the *Smad2/3* cKO organoids led to unopposed BMP/SMAD1/5 signaling, as has been described in other systems[32–35]. Gene ontology of downregulated genes showed that genes in the WNT signaling pathway were overrepresented (adj. $p = 4.79*10^{-4}$) (Fig. 5e, Supplementary Data 1), including genes such as Wnt family member 9a (*Wnt9a*) and the frizzled class receptor-1, −2, −7 and −10 (*Fzd1, Fzd3, Fzd7, Fzd10*) (Fig. 5e). Similar gene ontology groups were identified when we compared upregulated genes in control + A83-01 vs. control + vehicle organoids (Supplementary Data 1).

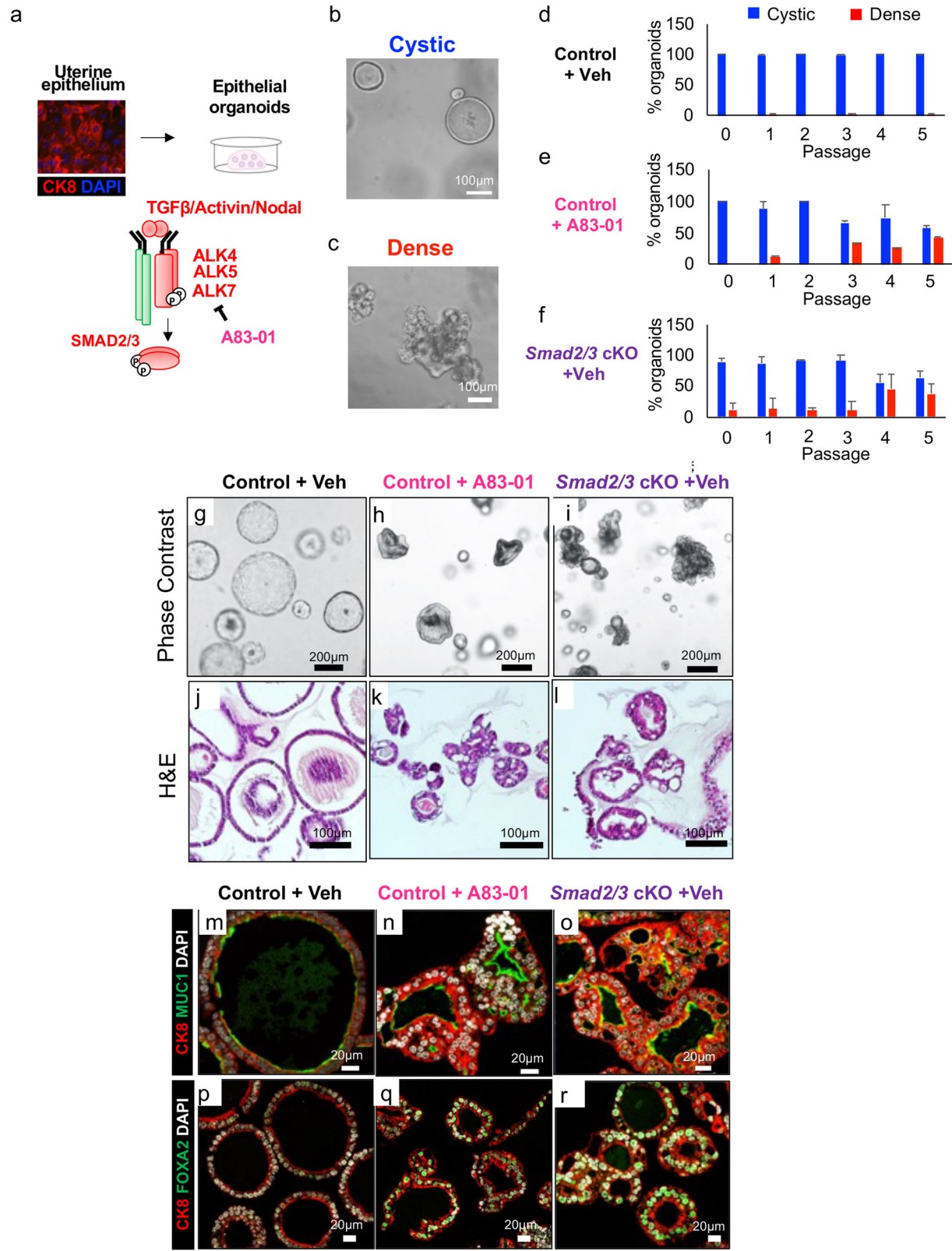

Differentially expressed genes involved in retinol metabolism (*Aldh1a1, Aldh1a2, Cyp26a1* and *Lrat*) and BMP signaling (*Id1* and *Id4*), were validated in a separate set of endometrial organoids using quantitative real time PCR (qPCR) and shown to be upregulated in the organoids from *Smad2/3* cKO mice (Fig. 5f). Hence, using endometrial epithelial organoids, we found that genetic or pharmacological suppression of SMAD2/3 signaling altered pathways that control retinoid metabolism, BMP and WNT signaling in endometrial organoids.

**Genome-wide SMAD4 binding in endometrial organoids reveals altered signaling response in *Smad2/3* cKO organoids.** Our genetic mouse models and organoid studies indicate that

**Fig. 4 Inhibition of SMAD2/3 signaling disrupts morphology and differentiation of endometrial epithelial organoids. a** Schematic showing the strategy used to isolate uterine epithelium from control and *Smad2/3* cKO mice for the culture of epithelial organoids. Organoids from control mice were grown in the presence or absence of the TGFβ receptor (ALK4/ALK5/ALK7) inhibitor, A83-01 for 5 passages. **b**, **c** Phase contrast images of cystic (**b**, round-shaped) or dense (**c**, lobular shaped) organoids. **d**–**f** Quantification of cystic vs. dense organoids across the three conditions, control + vehicle (**d**), control + A83-01 (**e**), and *Smad2/3* cKO (**f**). Values displayed as percentage of total organoids from $n = 3$ mice per group over 5 passages, mean ± SEM. **g**–**i** Phase contrast imaging of the organoids from control mice grown in the absence (**g**) or presence (**h**) of A83-01, and from *Smad2/3* cKO mice (**i**). **j**–**l** H&E stained cross sections of organoids from control mice (**j**), control mice treated with the A83-01 inhibitor (**k**), and from *Smad2/3* cKO mice (**l**). **m**–**r** Cross sections of endometrial organoids from control mice cultured in the absence (**m**, **p**) or presence of A83-01 (**n**, **q**) or from *Smad2/3* cKO mice (**o**, **r**). The organoids were immunostained with the epithelial cell marker antibody, cytokeratin 8 (CK8, red) and the mucin 1 antibody (**m**–**o**, MUC1, green), or with CK8 (red) and the glandular cell marker, FOXA2 (**p**–**r**, green). These experiments were performed in organoids derived from at least three mice per group.

perturbed TGFβ pathways control endometrial cell division and organization in the mouse uterus through the control of RA, WNT, and BMP-related gene expression. To investigate the molecular mechanism of TGFβ signaling at the chromatin level, we examined the genomic impact of conditional ablation of SMAD2/3 in the epithelium (Fig. 6). We performed Cleavage Under Targets and Release Using Nuclease (CUT&RUN)[36] of SMAD4 on organoids derived from *Smad2/3* cKO and control mice (Fig. 6a). SMAD4 is the common SMAD that is recruited to DNA by activated pSMAD2/3 or pSMAD1/5[37]. We hypothesized that SMAD4 binding in the epithelial organoids from control mice would be representative of both TGFβ/Activin/SMAD2/3/4 and BMP/SMAD1/5/4 events, while SMAD4 binding in the *Smad2/3* cKO mice would represent genomic binding events dictated only by BMP/SMAD1/5/4 signals.

We visualized the SMAD4 CUT&RUN binding sites over the transcription units as shown in Fig. 6b and found that SMAD4 signals are enriched near the transcription start sites in both control and *Smad2/3* cKO groups. Further genomic annotation confirmed the distribution of SMAD4 peaks is clustered toward the ±3 kb region surrounding promoters (Fig. 6c), which is consistent with the canonical role of SMAD4 as the common transcription factor facilitating TGFβ signal transduction[37]. As expected, organoids from *Smad2/3* cKO mice displayed fewer SMAD4 binding sites (Control 31080 vs. *Smad2/3* cKO 859).

We also analyzed the DNA motifs enriched in SMAD4 CUT&RUN peaks (Fig. 6d). We observed well-annotated GTCTG Smad binding elements ranked highly in our results. Motif sequences were also enriched for bZIP-Jun family transcription factors, validating the interaction of SMAD4 and Jun family proteins[38]. Interestingly, our analysis indicated that the DNA sequences representing Krüppel-like Factor (KLF) transcription factors are only enriched in the SMAD4 binding sites in the control but not *Smad2/3* cKO organoids, suggesting that regulatory networks of KLF and TGFβ pathways play an important role in maintaining the homeostasis in the endometrium.

We correlated SMAD4 binding events with changes in gene expression from our RNAseq analyes and found that 607 genes which had decreased expression in *Smad2/3* cKO vs. control organoids, could be classified as direct SMAD2/3 target genes (Fig. 6e and Supplementary Data 2). Alternately, 185 genes which had increased expression in *Smad2/3* cKO vs. control organoids and a SMAD4 binding event could be classified as SMAD1/5 target genes (Fig. 6e and Supplementary Data 2). Gene ontology analysis of the 4587 (FDR < 0.05) differentially bound peaks between control and *Smad2/3* cKO organoids confirmed that 4/10 top GO categories showed enrichment in TGFβ related pathways (Fig. 6e).

We also performed enrichment analysis of the 607 unique SMAD2/3 target genes and the 185 SMAD1/5 target genes to further characterize the gene-level differences between SMAD4 binding in control and *Smad2/3* cKO organoids (Supplementary Fig. 6 and Supplementary Data 2). These analyses showed that genes with SMAD4 binding sites in the control organoids regulate

pathways related to proteoglycan signaling, MAPK signaling and tight junction assembly (Supplementary Fig. 6). Genes with SMAD4 binding sites in the *Smad2/3* cKO organoids also not only control pathways related to MAPK signaling, but also display unique categories, such as cellular senescence and osteoclast differentiation (Supplementary Fig. 6). Thus, the gene expression programs directed by SMAD4 binding were different in control and *Smad2/3* cKO organoids. As an example of the differential regulation between control and *Smad2/3* cKO organoids, we demonstrated that the upstream promoter region of *Id3* showed SMAD4 enrichment in the *Smad2/3* cKO organoids when compared to control organoids, suggesting that the absence of TGFβ/Activin/SMAD2/3/4 signaling activates the BMP/SMAD1/5/4 axis (Fig. 6f).

**Ablation of SMAD2/3 signaling perturbs retinoid metabolism, BMP signaling and regeneration in the endometrium.** We identified that conditional ablation of SMAD2 and SMAD3 signaling in endometrial organoids increased expression of retinoid metabolism genes and BMP-regulated genes (Fig. 7). The ALDH1A1, ALDH1A2, and ALDH1A3 enzymes catalyze the oxidation of retinaldehyde into retinoic acid and are considered markers of adult stem cells in many tissues[39]. We verified their expression in uterine cross-sections from control and *Smad2/3* cKO mice and observed that ALDH1A1 and ALDH1A3 strongly localized to the crypts of the endometrial glands (Fig. 7a–d, i–l), while ALDH1A2 localized to the stroma of the endometrium with strong sub-epithelial expression (Fig. 7e–h). Similar expression patterns in the glandular crypts have been observed for ALDH1A1 and ALDH1A2 in the adult mouse uterus[40], while the neonatal uterine epithelium has been found to ubiquitously express ALDH1A1[41]. This dynamic pattern of expression (ubiquitous in the neonatal uterus vs. restricted to the endometrial crypts of adults) was previously reported for the leucine rich repeat containing G protein coupled receptor, LGR5, a marker of endometrial stem/progenitor cells in the endometrium[42]. At the mRNA level, we found that compared to control uterine epithelium, there was a trending increase in *Aldh1a1, Aldh1a2, Aldh1a3, Lrat* and *Rbp4* expression, and a significant increase in *Cyp26a1* expression in the uterine epithelium of *Smad2/3* cKO mice (Fig. 7u).

We also analyzed the expression of active pSMAD2 and pSMAD1/5 on uterine cross-sections from control and *Smad2/3* cKO mice. Strong pSMAD2 expression was localized to the uterine epithelium with some positive staining in the stromal compartment in control mice (Fig. 7m, n), while the uteri of *Smad2/3* cKO mice was negative for pSMAD2 in the epithelium with a few positive cells in the stromal compartment (Fig. 7o, p), indicating effective deletion by *Ltf*-cre. Expression of pSMAD1/5, on the other hand, showed weak epithelial and stromal expression in the control uterus, but strong expression in the epithelium of the *Smad2/3* cKO mice (Fig. 7q–t). This correlated with elevated expression of the canonical BMP-activated genes, *Id1, Id2, Id3,*

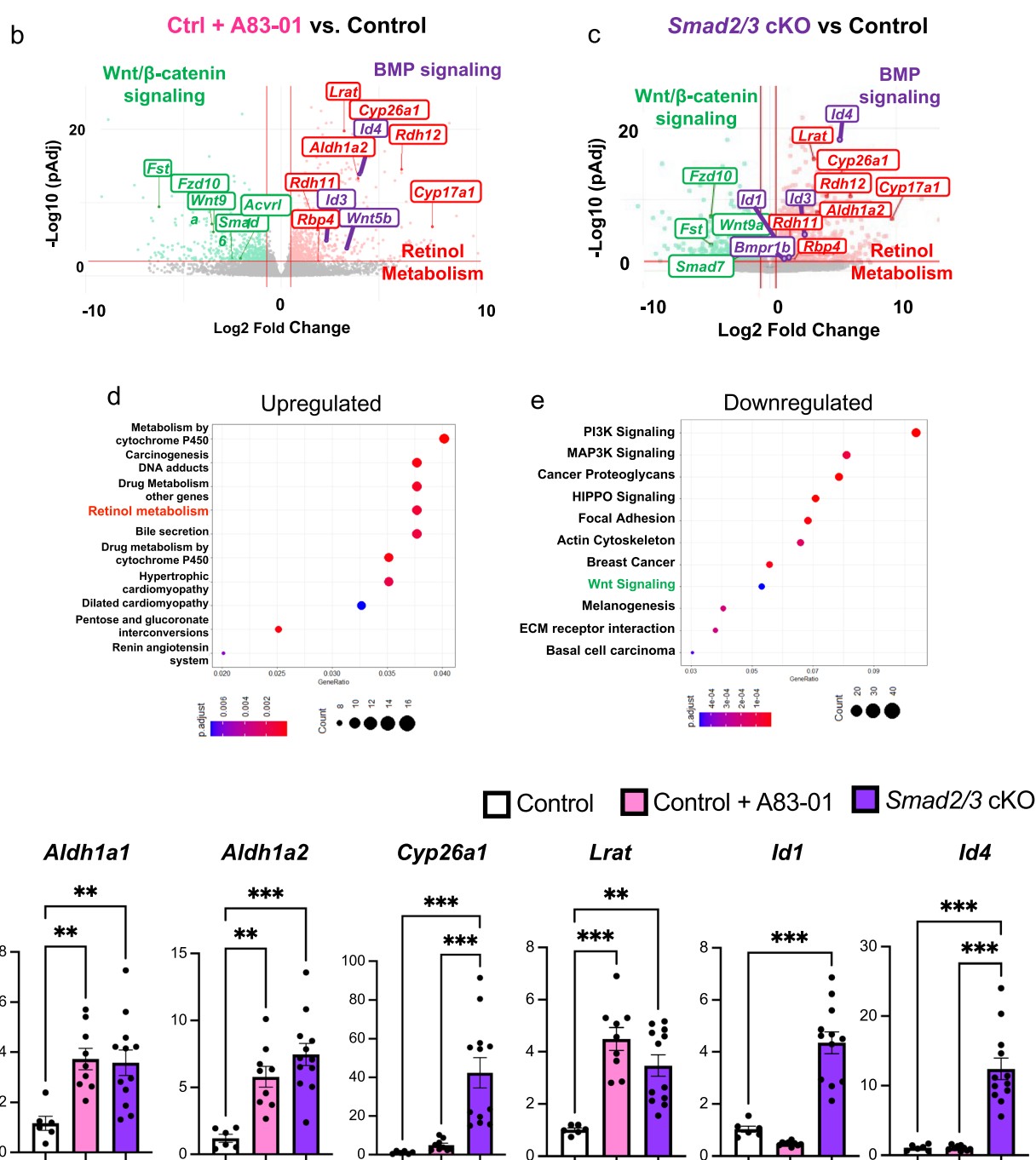

| | Ctrl + A83-01 vs. Ctrl | Smad2/3 cKO vs. Ctrl |
|---|---|---|
| > 1.4 FC, FDR < 0.01 | 569 | 912 |
| < 0.714 FC, FDR < 0.01 | 570 | 865 |

and *Id4* in the uterine epithelium of *Smad2/3* cKO uteri (Fig. 7u). These results suggest that the conditional SMAD2/3 inactivation in the mouse uterus leads to elevated BMP/SMAD1/5 signaling. This antagonistic mechanism between the TGFβ/SMAD2/3 and BMP/SMAD1/5 signaling pathways has been demonstrated in various other tissue systems and disease models[43–47], and is consistent with our results from the endometrial organoids.

## Discussion

Regeneration and differentiation in the endometrium are driven by stem cells that are directed to divide and regenerate throughout the reproductive lifespan[1,4,48]. In primates, it has been hypothesized that these stem cells reside in the deep basalis endometrium, where they aid in the rapid regeneration of the endometrium following menstruation[48,49]. However, recent

**Fig. 5 Gene expression profiling of endometrial organoids reveals that inhibition of SMAD2/3 signaling increases retinoid- and BMP-signaling pathways.** RNA-sequencing of the endometrial epithelial organoids was performed to identify the gene expression differences between control and *Smad2/3* cKO organoids. **a** Differentially expressed genes identified by RNAseq between the various organoid groups (Control + A83-01 vs. Control and *Smad2/3* cKO vs. Control). **b**, **c** Volcano plots highlighting gene-level differences identified by RNAseq between Control + A83-01 vs. Control (**b**) and *Smad2/3* cKO vs. Control (**c**) organoids. RNAseq data represent differentially expressed genes from four different mice per group, >1.4 fold or <0.714 fold change, FDR < 0.01. **d**, **e** Gene ontology analyses of overexpressed genes in *Smad2/3* cKO organoids indicates that "Retinol Metabolism" pathway genes are overrepresented in *Smad2/3* cKO organoids (**d**), while pathways related to "WNT/β-catenin" and are downregulated (**e**). **f** qPCR validation of differentially expressed genes identified by RNAseq in an independent set of organoids from Control, Control + A83-01, and *Smad2/3* cKO mice (n = 6, n = 9 and n = 12, per group, respectively). RNAseq was performed in organoids from four mice per group. Histograms represent mean ± SEM analyzed by an ordinary One Way ANOVA with Tukey multiple comparison post test.

studies indicate that stem cells may be located throughout the endometrium, suggesting a more efficient approach to ensure the rapid regeneration following menstruation[50–52]. The growth factors and signaling pathways that direct the regeneration or differentiation of these stem cells are not yet well-characterized. Studies in endometrial epithelial organoids have indicated that the WNT/β−catenin and Notch signaling pathways are critical for controlling the stem-like state of endometrial stem/progenitors[7,8,50]. Our results indicate that ligands of the TGFβ family signaling via the SMAD2 and SMAD3 transcription factors are also critical mediators of endometrial renewal and homeostasis by controlling RA, BMP, and WNT signaling pathways (Fig. 8). In addition, we identified that ALDH1A1 and ALDH1A3 are putative markers of endometrial stem cells located in the crypts of the endometrial glands. However, further lineage tracing experiments and mechanistic studies are required to classify them as a true stem cell population.

Although the mouse does not cyclically shed its endometrium through menstruation, it does undergo dynamic remodeling throughout the estrous cycle via endometrial resorption and in the post-partum phase, in which the entire endometrium is rapidly regenerated within 24–72 h[53–55]. During the post-partum period, endometrial repair in the mouse occurs via stromal-to-epithelial differentiation[54,56], epithelial cell migration[54], or recruitment of bone marrow-derived progenitor cells[57]. Various methods have been used to characterize the identity of mouse endometrial stem and progenitor cells, including label-retention[58] and lineage tracing with genetic markers[42,59–61]. This is an active area of investigation, and future studies will likely reveal the identity, location, and signals controlling the fate of endometrial stem cells.

Pharmacological inhibition of TGFβ signaling is required for the regenerative potential of human endometrial mesenchymal stem cells[5,6] and in human and mouse epithelial organoid cultures[7,8]. Similar to our findings in mouse epithelial organoids, sustained inhibition of TGFβ signaling with the ALK4/5/7 inhibitor resulted in elevated enrichment of retinoic acid signaling. These results indicate that the networks controlling endometrial regeneration are conserved between the endometrial stroma and epithelium. They also point to a critical role of retinoid signaling in the maintenance of stemness in the endometrium.

Our organoid studies also suggest a relationship between the TGFβ and WNT signaling pathways. Previous studies had shown that mouse endometrial organoids cultured under low WNT3a concentrations for 4 passages also developed the dense morphology that we observed[7]. This dense morphology could be restored to the round cystic morphology by increasing WNT3a concentration in the media. Given that these studies were performed in the presence of A83-01, and that our RNAseq studies identified a decrease in WNT signaling, it is likely that TGFβ activates WNT signaling to maintain the epithelial/progenitor cell state. Whether this signaling is occurring directly or indirectly via BMP or RA signaling remains to be determined.

In this study, we observed abnormal differentiation of the endometrium and transformation into endometrial cancer

following conditional ablation of the downstream effectors of TGFβ signaling, SMAD2 and SMAD3. In addition to TGFβ, other ligands such as activin, nodal, and growth differentiation factors can stimulate signaling through SMAD2/3[9]. Identifying the ligands that are signaling via SMAD2/3 to promote epithelial cell homeostasis will be critical to our understanding of endometrial cell regeneration and differentiation. This could help guide future targeted therapies, given the prevalence of ACVR2A K437 frameshift mutations in endometrial tumors (Supplementary Fig. 1i).

Receptors of the TGFβ signaling family that can activate SMAD2/3 signaling include the type 1 receptors, ALK4/ALK5/ALK7 and the type 2 receptors, ACVR2A, ACVR2B, and TGFBR2[9]. We previously observed that conditional deletion of the TGFβ type 1 receptor, TGFBR1/ALK5, directed endometrial cell regeneration in the post-partum phase in mice[19]. It has also been shown that TGFBR1 and TGFBR2 control uterine development and endometrial integrity using an *Amhr2*-cre conditional model[12,62]. Given that TGFβ is the major ligand activating SMAD2/3 via TGFBR1 and TGFBR2[63], it is likely that TGFβ plays a crucial role in endometrial regeneration. However, whether other ligands are implicated in this process remains to be identified.

We generated mouse models with single and double conditional inactivation of SMAD2 and SMAD3 using *Ltf*-cre, which is highly expressed in the uterine epithelium of adult mice[26]. Given that maximal *Ltf*-cre was observed at 60 days of age, we treated mice with estrogen at 21 days of age to induce cre recombination[26,64,65]. We found that while single *Smad2* cKO and *Smad3* cKO mice displayed normal uterine architecture, *Smad2/3* cKO mice developed tumors with lung metastases and died by ~9 months of age. We previously generated mice with conditional SMAD2/3 deletion using PR-cre ("SMAD2/3-PR-cre") and also observed development of metastatic endometrial cancer[18]. However, the onset of hyperplasia and tumor development was more rapid in the SMAD2/3-PR-cre mice than in the SMAD2/3-*Ltf*-cre model presented here. We also observed differences in expression of the glandular epithelial marker, FOXA2[66], where the SMAD2/3-PR-cre mouse line expressed higher levels of FOXA2 than age-matched controls but the SMAD2/3-*Ltf*-cre mice did not. This discrepancy in FOXA2 expression between the two mouse lines can be explained by the fact that the SMAD2/3-PR-cre mice developed endometrial hyperplasia much sooner (by 6 weeks of age) than the SMAD2/3-*Ltf*-cre mice (by 12 weeks of age), causing expression of FOXA2, a glandular marker, to appear more prominent than that of controls in the SMAD2/3-PRcre model. Furthermore, differences in the cell types expressing cre (stromal/epithelial in PR-cre vs. epithelial-specific in *Ltf*-cre), or the earlier onset of PR-cre recombinase activity compared to *Ltf*-cre, could also account for the differences noted in FOXA2 expression.

To determine whether the tumors were estrogen-dependent, we ovariectomized control and *Smad2/3* cKO mice and treated them with either a placebo or E2-releasing pellet. We found that

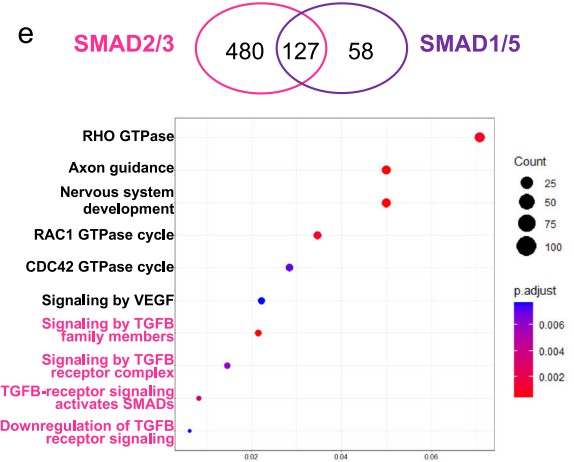

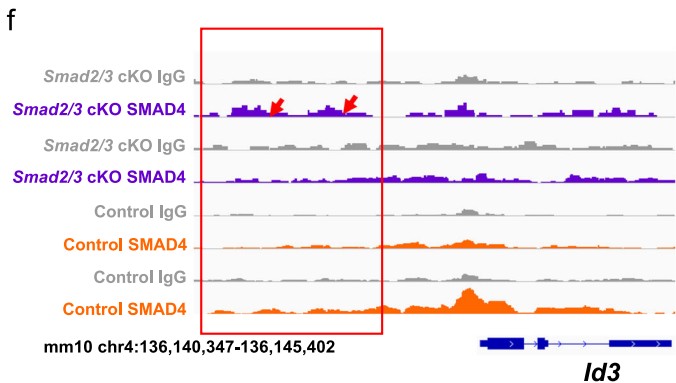

| Sequence | p-value | Name | Ranks | Geno |
|---|---|---|---|---|
| GGCCGTCTGG | $10^{-17}$-$10^{-30}$ | SMAD4 | 1-3 | *Smad2/3* cKO |
| GATGACTCATCC | $10^{-7}$-$10^{-11}$ | bZIP/Jun/AP1 | 4-10 | |
| GATGACTCATC | $10^{-668}$-$10^{-801}$ | bZIP/Fra1/Jun | 1-9,13 | Control |
| CTGTCTGG | $10^{-428}$-$10^{-509}$ | SMAD4 | 10-12 | |
| TGGGCTGTGGC | $10^{-74}$-$10^{-108}$ | KLF | 14,16, 23 | |

only *Smad2/3* cKO mice with E2 pellets developed tumors. Because *Ltf*-cre activity is E2-dependent, we also tested the effect of E2 on tumor development independently of *Ltf*-cre by using adenoviral-mediated deletion of SMAD2/3. We found that only *Smad2*flox/flox;*Smad3*flox/flox mice with intrauterine Ad-cre injection and E2-pellets developed tumors, confirming that E2 is necessary for tumor development in *Smad2/3* cKO mice.

*Ltf*-cre activity is also found in male reproductive tissues and in the esophagus, and in few cells of non-reproductive tissues[26]. Our survival analyses showed that most mice that perished displayed uterine masses; however, 4/18 mice did not, suggesting that death due to the deletion of SMAD2/3 in non-uterine tissues occurred in a few mice. This could be expected, given the relevance of TGFβ signaling in other organs[10]. To ensure that the lung

**Fig. 6 Analysis of SMAD4 binding in endometrial organoids reveals differential binding across the genome. a** Diagram outlining the procedures used to identify SMAD4-bound genes in organoids from Control and *Smad2/3* cKO mice using CUT & RUN. **b** Heatplot showing the SMAD4 signal distribution across the transcriptional start site (TSS) and transcriptional end site (TES) in Control and *Smad2/3* cKO organoids. As expected, the SMAD4 signal in the *Smad2/3* cKO organoids was decreased when compared to the SMAD4 signal in Control organoids. **c** Feature distribution comparison between the SMAD4 binding sites in Control and *Smad2/3* cKO organoids. **d** Motif sequence analyses in the SMAD4-bound regions in Control and *Smad2/3* cKO organoids. **e** Differentially bound SMAD4 genes in Control (representing SMAD2/3 targets) and *Smad2/3* cKO organoids (representing SMAD1/5 targets) and the gene ontology analysis of the differentially bound genes in Control organoids. **f** Genome track screenshot showing increased SMAD4 enrichment in the upstream promoter region of the BMP-target gene, *Id3*, in *Smad2/3* cKO organoids when compared to Control organoids. CUT & RUN experiments were performed in the organoids from >3 mice per genotype, analyzed and sequenced as duplicates.

nodules were of uterine origin and did not arise due to loss of SMAD2/3 in the lungs, we demonstrated that these metastatic lesions did not express TTF1, a lung-specific marker, and did express ERα.

Using SMAD4 CUT&RUN in endometrial organoids, we confirmed that the binding of SMAD4 was enriched near transcriptional start sites and that SMAD binding elements were enriched in the DNA sequences bound by SMAD4. Our SMAD4 CUT&RUN data also revealed key differences in the binding patterns of SMAD4 in the organoids from control and *Smad2/3* cKO mice. Firstly, we identified fewer SMAD4 binding sites in the *Smad2/3* cKO mice as well as decreased SMAD4 binding signal in the organoids from *Smad2/3* cKO. This was expected, given that the binding of SMAD4 in the organoids derived from *Smad2/3* cKO mice is driven only by the action of SMAD1/5. We also found that the while SMAD binding elements were enriched in both the binding sites from control and *Smad2/3* cKO organoids, KLF binding elements were enriched only in the organoids from *Smad2/3* cKO organoids. This suggests that the absence of SMAD2/3 alters SMAD4 transcriptional activity, leading to its enrichment in KLF-enriched motifs. Our studies in the mouse uterus previously identified SMAD4 enrichment in the promoter region of KLF transcription factor 15 (*Klf15*), a key factor driving the anti-proliferative action of E2 in the uterus[11,67]. Thus, exploring additional interactions between SMAD4 and the KLF family of proteins may reveal clues about the shared activity of these pathways in the uterus.

Uncovering the pathways that underlie normal endometrial homeostasis and regeneration is critical for designing improved therapies that target endometrial pathologies. This is especially true for endometrial cancer, which displays a rapidly rising incidence in the United States and world-wide[68,69]. Approximately 65,000 women will be diagnosed with endometrial adenocarcinoma in 2022, leading to ~12,550 deaths[70]. Therefore, shedding light on the factors that control endometrial cell regeneration and differentiation will be key to improving gynecological health.

## Methods

**Animal ethics statement.** All mouse handling and experimental studies were performed under protocols approved by the Institutional Animal Care and Use Committee of Baylor College of Medicine and guidelines established by the NIH Guide for the Care and Use of Laboratory Animals. All the mice were housed under standard conditions of a 12 h light/dark cycle in a vivarium with controlled ambient temperature (70 °C ± 2 °C and 20–70% relative humidity).

**Generation of *Smad2/3* cKO mouse lines.** The Smad2/3 cKO mouse line was generated by using the Lactoferrin-iCre (*Ltf*-cre) LoxP system[26]. Briefly, *Ltf*-cre mice were bred to *Smad2*flox/flox;*Smad3*flox/flox mice to generate *Smad2*flox/flox; *Smad3*flox/flox-*Ltf*-cre/+ males[15,25]. *Smad2*flox/flox;*Smad3*flox/flox females were mated with *Smad2*flox/flox; *Smad3*flox/flox -*Ltf*-cre/+ males to produce female offspring with the genotypes, *Smad2*flox/flox; *Smad3*flox/flox (control) and *Smad2*flox/flox; *Smad3*flox/flox-*Ltf*-cre/+ (*Smad2/3* cKO) for the studies. Mouse genotyping was performed by using DNA extracted from 2 to 3 mm tail snips that were digested in 200 μl of 50 mM NaOH at 95 °C for 30 min, followed by addition 100 μl of 1 M Tris-HCl, pH 8 and centrifugation at maximum speed for 5 min. The isolated DNA (1–2 μL) was PCR amplified using the primer sequences listed in Supplementary Table 2.

**Surgeries and hormone treatments.** Mice were administered with 2 doses of 100 ng of Estradiol (Sigma, dissolved in sesame oil) at the time of weaning (~21 and 22 days of age) to induce *Ltf*-cre activity[26]. All surgeries and hormone treatments were performed following IACUC-approved procedures. Six- to eight-week-old control and *Smad2/3* cKO mice were ovariectomized and implanted (s.c.) with placebo or estradiol pellets (17β-ESTRADIOL, 0.025 mg, 90 days, Innovative Research of America, NE-121).

**Deletion of SMAD2 and SMAD3 by adenoviral-cre intrauterine injections.** Adenoviral-Empty and adenoviral-Cre were obtained from Advanced Technology Core at Baylor College of Medicine. Adult female *Smad2*flox/flox; *Smad3*flox/flox mice were anesthetized with isoflurane and their ovaries were removed. Each uterine horn was then injected with 30 μL of either adenoviral-Empty or adenoviral-Cre (total $1.05 \times 10^8$ pfu)[29,30]. Female mice were implanted with a placebo or estradiol pellet (17β-ESTRADIOL, 0.025 mg, 90 days, Innovative Research of America, NE-121) and collected after 90 days.

**Tissue collection for nucleic acid and protein analyses.** Tissues were harvested immediately after euthanasia and fixed in 10% Formalin (Sigma) overnight at room temperature. The tissues were then switched to 70% ethanol and submitted for paraffin processing and embedding at the Human Tissue Acquisition and Pathology Core at Baylor College of Medicine. Tissues intended for protein or mRNA analysis were harvested and immediately frozen in dry ice until further extraction.

**Extraction of mRNA for analysis of gene expression.** For mRNA extraction of isolated epithelium, frozen tissues were lysed with RLT buffer and processed following manufacturer's procedures (RNeasy Micro Kit, Qiagen) using the DNase on column digest. Approximately 1 μg of mRNA was reverse transcribed into cDNA using qScript cDNA Supermix (Quanta Bio, 101414-106) and amplified using specific primers (listed in Supplementary Table 2). Primers were amplified using 2X SYBR Green Reagent (Life Technologies, 4364346) using a BioRad CFX384 Touch Real Time PCR Detection System. Data were analyzed using relative quantification, ΔΔCt[71].

**Histological analyses and imaging.** Tissues were then sectioned into 5 μm thick sections and used for histological stains or immunostaining with antibodies listed in Supplementary Table 3. For immunostaining and IHC, tissues were subjected to antigen retrieval in 10 mM Citrate Buffer with 0.5% Tween, pH 6.0 in a microwave for 20 min. Tissues were then incubated overnight at 4° with primary antibodies resuspended in 3% BSA followed by incubation with fluorophore-conjugated secondary antibodies (Alexa-Fluor-488 or Alexa-Fluor-594, Invitrogen) and mounted with Vectashield mounting medium (Vector Labs). For IHC, sections were incubated with biotinylated secondary antibodies followed by incubation with a signal amplification avidin/biotin complex (Vector Labs, PK-6100) and developed with DAB peroxidase substrate (Vector Labs, SK-4100). Sections were counterstained with Hematoxylin (Sigma), dehydrated, and mounted using Permount mounting medium (VWR). Peroxidase-labeled and H&E-stained slides were imaged using an Olympus BX41 light microscope and images were captured using a Nikon DS-Fi2. Fluorescently labeled slides were imaged at the Optical Imaging and Vital Microscopy Core Facility Laboratory at Baylor College of Medicine using an LSM880 confocal microscope.

**Epithelial cell isolation from the mouse uterus.** Isolation of mouse uterine epithelium was performed by incubating 2–3 mm uterine fragments in 1% Trypsin (Sigma, T1426) dissolved in Hank's Balanced Salt Solution for 60 min at 37° C followed by mechanical separation of the epithelial sheets from the uterus under a dissection microscope[72]. For mRNA or protein extraction, the uterine epithelial cells were immediately frozen in dry ice followed by downstream analysis. To generate endometrial organoids, the epithelium was mechanically separated from the uterus and further digested into single cells using a brief 3–5-minute mechanical dissociation in 2.5 mg/ml Collagenase (Sigma) and 2 μg/ml DNase. Once single cells were obtained, the epithelium was encapsulated in Matrigel as described in the section below.

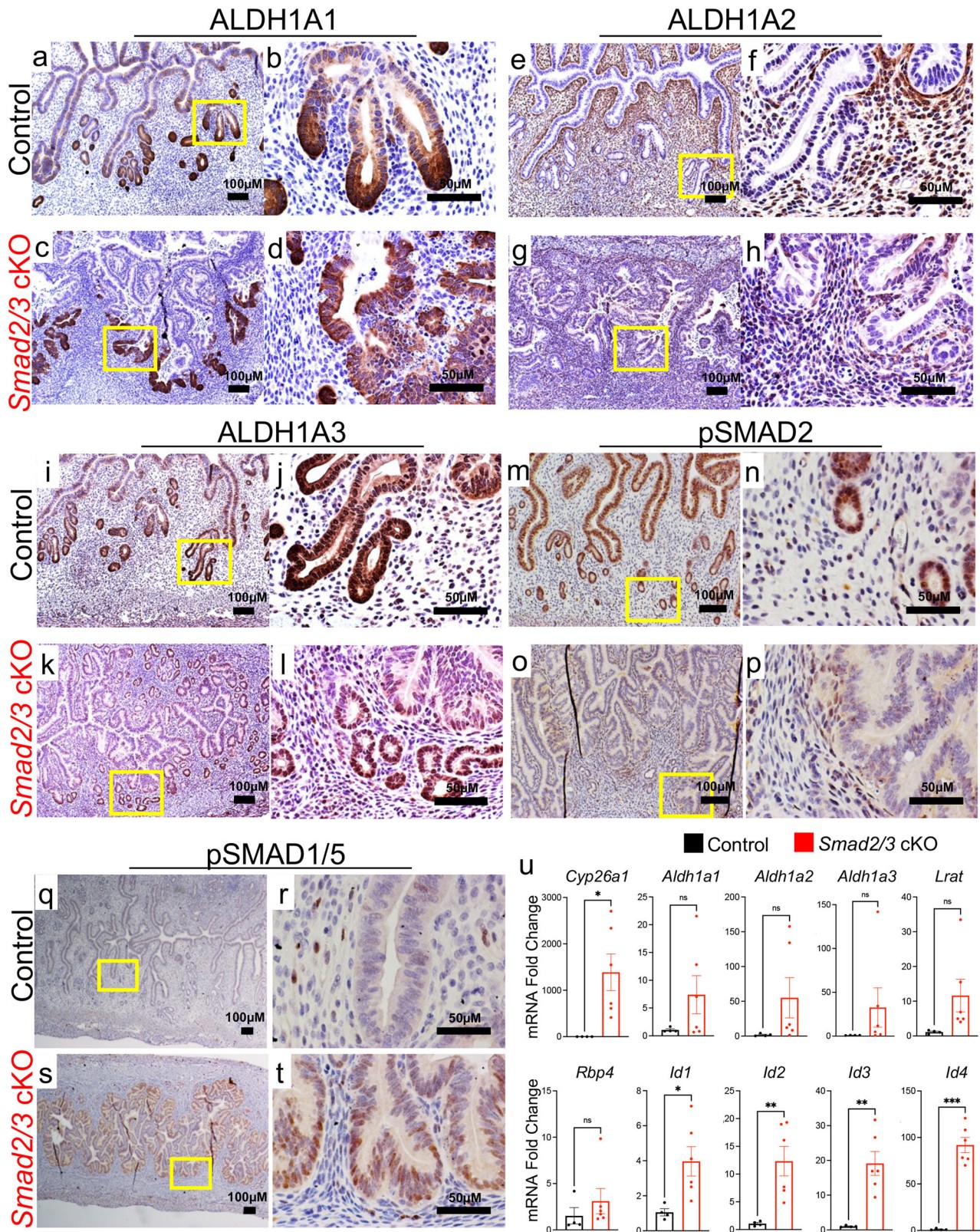

**Generation of endometrial organoids**. Endometrial organoids were established following the methods and culture conditions described by Boretto et al.[7], with minor changes[72]. Specifically, once the isolated epithelium in a single cell suspension were obtained as described above, the cell pellets were encapsulated in ice-cold Matrigel (Corning, 354230) at a 1:20 ratio of cell pellet volume:Matrigel (i.e., a 10 μl cell pellet was resuspended in 200 μl Matrigel), and allowed to solidify at room temperature for 10 min in a 1.5 ml Eppendorff tube. Once the Matrigel was solid, a wide-bore 200 μl pipette was used to dispense 3–25 μl domes into a 12-well

plate. The domes were allowed to settle for 10 min in the 37 °C tissue culture incubator and were then overlayed with 750 μl of Organoid Medium. Organoid medium was comprised of the following ingredients: Advanced DMEM/F12 (Life Technologies, 12634010), 1X N2 Supplement (Life Technologies, 17502048), 1X B-27 minus vitamin A (Life Technologies, 12587010), 100 μg/ml Primocin (Invivogen, ant-pm-1), 1.25mM N-Acetyl-L-cysteine (Sigma, A9165), 2mM L-Glutamine (Life Technologies, 25030024), 10 nM Nicotinamide (Sigma, N0636), 50 ng/ml recombinant human EGF (Peprotech, AF-100-15), recombinant human

**Fig. 7 Detection of retinoid- and BMP-signaling pathways in control and *Smad2/3* cKO mice. a–d** Cross-sections from 17-week-old control (**a**, **b**) and *Smad2/3* cKO (**c**, **d**) mice stained with ALDH1A1 antibody. ALDH1A1 is enriched in the crypts of the mouse endometrial glands. **e–h** ALDH1A2 IHC in the uteri of control (**e**, **f**) and *Smad2/3* cKO mice (**g**, **h**) ALHD1A2 is localized to the subepithelial stromal compartment. **i–l** ALDH1A3 IHC in control (**I**, **j**) and *Smad2/3* cKO (**k**, **l**) mice shows enrichment in the crypts of endometrial glands. **m–p** pSMAD2 IHC in control (**m**, **n**) and *Smad2/3* cKO (**o**, **p**) uterine cross-sections. Decreased pSMAD2 is observed in *Smad2/3* cKO mice. **q–t** pSMAD1/5 IHC in the uterine cross-sections of control (**q**, **r**) and *Smad2/3* cKO (**s**, **t**) mice shows increased pSMAD1/5 reactivity in the uteri of *Smad2/3* cKO mice. IHC experiments were performed in >3 mice per genotype. **u** Quantitative PCR analysis of uterine epithelium from control (*n* = 4) and *Smad2/3* cKO (*n* = 6) for genes involved in retinoid signaling (*Cyp26a1, Aldh1a1, -1a2, -1a3, Lrat*, and *Rbp4*) or BMP signaling (*Id1, Id2, Id3, Id4*). Histograms represent mean ± SEM analyzed by an unpaired two-tailed *t*-test.

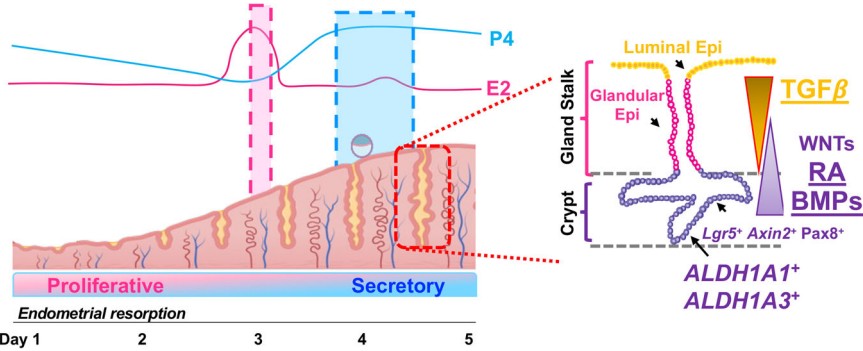

**Fig. 8 Schematic showing the effect of TGFβ signaling in endometrial cell regeneration and differentiation.** Diagram indicating the dynamic remodeling of the endometrium throughout the estrous cycle, transitioning from a proliferative to a secretory state under the control of the steroid hormones, estrogen (E2) and progesterone (P4). The regenerative potential of the endometrium is controlled by the presence of endometrial *Lgr5*+, *Axin2*+, and *Pax8*+ stem cells, likely in the crypts of the uterine glands, with growth factors such as WNTs, controlling differentiation. Our results indicate that *Aldh1a1*+ and *Aldh1a3*+ cells are putative endometrial stem cells in the uterine glands that are controlled by TGFβ, BMP and retinoic acid (RA) signaling.

FGF-10 (Peprotech, 100-26), recombinant human HGF (Peprotech, 100-39), 10% of conditioned WNT3a medium, 10% conditioned R-Spondin medium, and 10% conditioned Noggin medium (obtained from the Center for Digestive Diseases Core Facility at Baylor College of Medicine). Media was prepared in the presence or absence of 500 nM A83-01 (Tocris, 2939), then sterile filtered and stored at 4 °C. Organoids were passaged by resuspending in ice-cold Advanced DMEM and centrifuged at 600 × *g* for 5 min for a total of three times. Organoids were mechanically dissociated after each centrifugation step by resuspending in 100 µl of Advanced DMEM, pipetting ~100 times through a 200 µl pipette tip, followed by addition of 5 ml of ice-cold Advanced DMEM. Once the Matrigel was visibly separated from the organoids, excess Matrigel was removed, and the organoids were resuspended in sufficient Matrigel to split the organoids 1:3 or 1:4 ratio.

**Analysis of endometrial organoids by RNA sequencing**. Total mRNA was extracted from the endometrial organoids of 3–4 different mice per condition (3-control; 4-control + A83-01; 4- *Smad2/3* cKO) using the DirectZol kit from Zymo. RNA was ensured to have a high-quality RIN score and subjected to library preparation and sequencing using the Ultra-Low Input Library Preparation Kit (SMART Seq v4, Takara, Inc). Next Generation Sequencing was used to obtain ~20 million reads per sample using the Illumina Platform PE150 (Novogene, Inc). Reads were filtered, trimmed, and aligned to the mouse genome (build GRCm39) using Salmon 1.4.0. Differentially expressed genes were calculated using DEseq2 (version 1.32.0) with fold chance >1.4 and <0.714 and adjusted *p* value < 0.01 and visualized with ggplot2 (version 3.3.5). Gene ontologies of genes classified to be up- or downregulated were obtained using Sigterms v1[73], adjusted p-values and visualizations were created using enrichplot (R package version 1.16.1)[74]. Sequencing data are available in the Gene Expression Omnibus (GSE212475).

**Western blotting**. Isolated epithelium from control and *Smad2/3* cKO mice was frozen on dry ice and stored at −80 °C until lysis. Protein extraction was performed by lysing the epithelial tissue in T-PER (Pierce) supplemented with phosphatase and protease inhibitor. The tissue was lysed by vigorous vortexing (three times, for 30 s) with intermittent 5 min incubations on ice (3 min). Tissues were cleared by centrifugation at 14,000 *rpm* for 30 min at 4 °C in a tabletop centrifuge. The lysate was transferred to a clean tube, and protein concentration was determined using a BCA protein assay (Pierce). Approximately 20 µg of protein were subjected to SDS-PAGE on 4–16% polyacrylamide gels (Invitrogen) and transferred to nitrocellulose membranes using a semi-dry transfer system (Transblot Turbo, BioRad). The membranes were blocked in 5% milk in TBST and probed with SMAD2, SMAD3 and GAPDH antibodies (antibody information is listed in Supplementary Table 3). The uncropped blots are provided in Supplementary Fig. 2.

**SMAD4 genome-wide binding analysis using CUT&RUN**. Mouse endometrial organoids were digested with 0.25% Trypsin for 10 min at 37 °C to get single cell suspension. Next, CUT&RUN procedure largely follows a previous protocol[36]. Briefly, around 500,000 cells were used per reaction and duplicates were used for each genotype. The cells were washed twice with 1 ml wash buffer (20 mM HEPES pH = 7.5, 150 mM NaCl, 0.5 mM Spermidine, 1X Roche complete protease inhibitor). 10 µl of concanavalin-coated beads (Bangs Labs BP531) were washed twice in Bead Activation Buffer (20 mM HEPES pH = 7.9, 10 mM KCl, 1 mM CaCl₂, 1 mM MnCl₂) for each reaction. Then, beads were added to cell resuspension and incubated for 10 min (min) at room temperature. After incubation, bead-cell complexes were resuspended in 100 µl Antibody Buffer (wash buffer + 0.01% digitonin + 2 mM EDTA) per reaction. 0.678 µg of IgG (Sigma) and SMAD4 (Abcam, ab40759) antibodies were added to each group respectively. After overnight incubation at 4 °C, bead-cell complexes were washed twice with 200 µl cold Dig-Wash buffer (Wash buffer + 0.01% digitonin) and resuspended in 50 µl cold Dig-Wash buffer with 1 µl pAG-MNase (EpiCypher, 15–1016) per reaction. After incubation at room temperature for 10 min, bead-cell complexes were washed twice with 200 µl cold Dig-Wash buffer and resuspended in 50 µl cold Dig-Wash buffer, then 1 µl 100 mM CaCl₂ was added to each reaction. The mixture was incubated at 4 °C for 2 h and the reaction was stopped by adding 50 µl Stop Buffer (340 mM NaCl, 20 mM EDTA, 4 mM EGTA, 0.05% Digitonin, 100 µg/mL RNase A, 50 mg/mL glycogen, 0.5 ng E. coli DNA Spike-in (EpiCypher 18–1401)) and incubated at 37 °C for 10 min. The supernatant was collected and subjected to DNA purification with phenol-chloroform and ethanol precipitation. Sequencing libraries were prepared using NEBNext Ultra II DNA Library Prep Kit (NEB E7645) following manufacturer's protocol.

Paired-end 150 bp sequencing was performed on a NEXTSeq550 (Illumina) platform. Raw data were de-multiplexed by bcl2fastq v2.20 with fastqc for quality control. Clean reads were mapped to reference genome mm10 by Bowtie2 (v2.2.7), with parameters of --end-to-end --very-sensitive --no-mixed --no-discordant --phred33 -I 10 -X 700. For Spike-in mapping, reads were mapped to E. coli genome U00096.3. Duplicated reads were removed, and only uniquely mapped reads were kept. Spike-in normalization was achieved through multiply primary genome coverage by scale factor (100000 / fragments mapped to E. coli genome). CUT&RUN peaks were called by MACS2 (v2.1.0) with the parameters of -f BAM -q 0.1 -n. Track visualization was done by bedGraphToBigWig[75], bigwig files were imported to Integrative Genomics Viewer for visualization. For peak annotation, common peaks between duplicates were identified with 'mergePeaks' function in homer v4.11 and then genomic annotation was added by ChIPseeker[76]. Motif analysis was conducted through HOMER v4.11 on the merged peaks with parameter set as findMotifsGenome.pl mm10 -size 200 –mask. Integration of SMAD4 peaks with differentially expressed genes from the RNAseq analysis was performed using DiffBind with FDR < 0.05[77]. Sequencing data are available in the Gene Expression Omnibus (GSE212474).

**Analysis of TGFβ related mutations from the cBioPortal database**. A dataset consisting of 6 independent studies, 902 samples, and 894 patients was profiled for the presence of mutations related to the TGFβ signaling pathway. The dataset was obtained from the cBioPortal[20,21] and can be accessed at the following link: https://bit.ly/3Slqiw4.

**Statistics and reproducibility**. Analyses were performed in >3 biological and technical replicates using Excel or GraphPad Prism. Figure 1b, c were analyzed an unpaired two-tailed $t$-test to determine differences between the means of *Smad2* and *Smad3* expression in the uterine tissues of control and *Smad2/3* cKO mice. Figure 5f was analyzed by an ordinary One Way ANOVA with Tukey multiple comparison post-test to identify the differences between the means of each group of organoids. Figure 7u was analyzed by an unpaired two-tailed $t$-test to identify the differences between the means of genes expressed in the uterus of controls vs. *Smad2/3* cKO mice. Schematic diagrams in Figs. 6, 8 and Supplementary Fig. 5 were generated using BioRender.

**Reporting summary**. Further information on research design is available in the Nature Portfolio Reporting Summary linked to this article.

## Data availability

Sequencing analyses are freely available and deposited in the Gene Expression Omnibus under accession number GSE212477 superseries or by request from the communicating author. Source data behind the graphs in the paper are included in Supplementary Data 3. All other data are available from the corresponding author (or other sources, as applicable) on reasonable request.

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

## Acknowledgements

We are grateful to Dr. Martin M. Matzuk (M.M.M.) for his gracious support and guidance on this project and to Bradly Ramirez for analysis of the endometrial organoids. Studies were supported by *Eunice Kennedy Shriver* National Institute of Child Health and Human Development grants R00-HD096057 (D.M.), R01-HD105800 (D.M.), R01-HD032067 (M.M.M.) and R01-HD110038 (M.M.M.), and by NCI- P30 Cancer Center Support Grant (NCI-CA125123). Diana Monsivais, Ph.D. holds a Next Gen Pregnancy Award (NGP10125) from the Burroughs Wellcome Fund. Z.T. is a CPRIT scholar in cancer research and Z.T. thanks the CPRIT for research funding support (RR220039).

## Author contributions

Study conception and design: M.L.K., S.T., Z.L., D.M. Performed experiment or data collection: M.L.K., S.T., Z.L., P.J., F.Y., F.C., S.E.P., D.I.C., R.P.M., P.D.C., M.M.I., C.J.C., Z.T., D.M. Computation and statistical analysis: M.L.K., S.T., Z.L., P.J., F.Y., F.C., C.J.C., Z.T., D.M. Data interpretation and analysis: M.L.K., S.T., Z.L., P.J., F.Y., S.E.P., R.P.M., P.D.C., M.M.I., C.J.C., Z.T., D.M. Writing, reviewing and editing: All. Supervision: D.M.

## Competing interests

The authors declare no competing interests.
