## [Peer Review File · Communications Biology]

Reviewers' comments:

Reviewer #1 (Remarks to the Author):

The proposed manuscript presents interesting results concerning the role of Smad2/3 signaling in uterine epithelium cell homeostasis and regeneration. Based on pubmed search, the presented problem is novel and original. Obtained results are of great potential to interest the scientific community focused on the impact of TGF beta signaling on endometrial pathologies. The conclusions are precise and adequate to the presented results. According to my understanding and expertise, all performed experiments are well-designed and executed. In all experiments, the appropriate controls were applied. Moreover, the authors provide fully detailed information, which makes the whole study easily reproduce, the authors provided suitable references where needed. The presented study is supplemented with all necessary figures and tables, as well as supplementary material. In the discussion section, the authors present and discuss any potential limitations or need for further elucidation. Obtained results are well discussed with to date state of knowledge in the field.

Reviewer #2 (Remarks to the Author):

In the present study authored by Kriseman and collaborators, the role of SMAD2/3 in endometrial carcinogenesis is studied using an inducible lactoferrin-inducible knock-out mouse model. The manuscript clearly shows two different parts: the first one in which the endometrial phenotype of double SMAD3/3 KO is analyzed in vivo (Figure 1 to 5) and the second part, in which the authors address the molecular mechanisms of SMAD2/3 in endometrial carcinogenesis using endometrial organoids (Figure 6 to 8).

Major points:

The results presented in the first part of the study (Figures 1 to 5) are confirmatory results from those that have been previously published in a study authored by the same first author (Kriseman, M. et al. Uterine double-conditional inactivation of Smad2 and Smad3 in mice causes endometrial dysregulation, infertility, and uterine cancer. Proc Natl Acad Sci U S A 116, 3873-3882 (2019). The main difference between the present study and Kriseman et al., 2019 is the Cre mice used. The Progesterone-Cre used in the published study is no specific of endometrial epithelial cells whilst the one used here is specific of endometrial epithelial cells. However, the results are extremely similar to the previously reported ones and therefore, they do not provide further knowledge about the role of SMAD2/3 in endometrial cancer. Please, see the comparison between the Figures of the present manuscript and those published in Kriseman et al., 2019:

-Figure 3A-B and Figure 3K-L (E-cadherin, SMA immunostaining) results similar to those reported in Figure 2.

-Figure 3C-F and Figure 3M-P: results similar to those reported in Figure 4E-L (Progesterone Receptor and FOXA2 immunostaining).

-Figure 4A-D results are similar to those reported in Figure 5B-C (endometrium) and Figure F-G (lungs)

-Figure 4E-F results are similar to those reported in Figure I-L

-Kaplan-Meyer in Figure 4O is similar to that shown in Figure 5M.

-Figure 5A-B, Figure 5E-F are similar to Figure 6A-B.

We also found some apparent discrepancies between the results presented in Kriseman et al., 2019 and the present study (may be caused by different Cre mice used) :

Present study :

"...Similar to the 12-week timepoint, there was no observed change in FOXA2 immunoreactivity between Smad2/3 cKO and control mice at 6 months of age (Figure 3Q-T)."

Kriseman et al., 2019:

« Compared with controls (Fig. 4I and J), disordered uterine glandular structures in the Smad2/3cKO mice were visualized by immunohistochemistry of the glandular marker FOXA2 (Fig. 4K and L), with a corresponding increase in Foxa2 gene expression in the Smad2/3cKO mice relative to the controls (Fig. 4D) »

Other comments:

Figure 2E-G: an immunohistochemistry with p-SMAD3 or p-SMAD2/3 should be performed.

Regarding the second part of the manuscript, Figure 8 shows an scheme of the putative roles of TGF in the endometrium, leaving only Figures 6 and 7 as the only ones showing experimental data experimental. In Figure 6, the authors show an increase of retinol metabolism and downregulation of Wnt signaling identified by RNA-seq experiments. Although the results are potentially interesting, there is no functional assessment of the role these pathways on endometrial organoids.

Minor points:

Figure 1 is a c-bioportal search showing no experimental-based data. This figure should be moved to a supplementary material.

Figures 5E-H are not cited in the text.

Reviewer #1 (Remarks to the Author):

The proposed manuscript presents interesting results concerning the role of Smad2/3 signaling in uterine epithelium cell homeostasis and regeneration. Based on pubmed search, the presented problem is novel and original. Obtained results are of great potential to interest the scientific community focused on the impact of TGF beta signaling on endometrial pathologies. The conclusions are precise and adequate to the presented results. According to my understanding and expertise, all performed experiments are well-designed and executed. In all experiments, the appropriate controls were applied. Moreover, the authors provide fully detailed information, which makes the whole study easily reproduce, the authors provided suitable references where needed. The presented study is supplemented with all necessary figures and tables, as well as supplementary material. In the discussion section, the authors present and discuss any potential limitations or need for further elucidation. Obtained results are well discussed with to date state of knowledge in the field.

Thank you for your thorough review, comments, and positive opinion of our manuscript.

Reviewer #2 (Remarks to the Author):

In the present study authored by Kriseman and collaborators, the role of SMAD2/3 in endometrial carcinogenesis is studied using an inducible lactoferrin-inducible knock-out mouse model. The manuscript clearly shows two different parts: the first one in which the endometrial phenotype of double SMAD3/3 KO is analyzed in vivo (Figure 1 to 5) and the second part, in which the authors address the molecular mechanisms of SMAD2/3 in endometrial carcinogenesis using endometrial organoids (Figure 6 to 8).

Major points:

1. The results presented in the first part of the study (Figures 1 to 5) are confirmatory results from those that have been previously published in a study authored by the same first author (Kriseman, M. et al. Uterine double-conditional inactivation of Smad2 and Smad3 in mice causes endometrial dysregulation, infertility, and uterine cancer. Proc Natl Acad Sci U S A 116, 3873-3882 (2019). The main difference between the present study and Kriseman et al., 2019 is the Cre mice used. The Progesterone-Cre used in the published study is no specific of endometrial epithelial cells whilst the one used here is specific of endometrial epithelial cells. However, the results are extremely similar to the previously reported ones and therefore, they do not provide further knowledge about the role of SMAD2/3 in endometrial cancer.

Please, see the comparison between the Figures of the present manuscript and those published in Kriseman et al., 2019:

- Figure 3A-B and Figure 3K-L (E-cadherin, SMA immunostaining) results similar to those reported in Figure 2.
- Figure 3C-F and Figure 3M-P: results similar to those reported in Figure 4E-L (Progesterone Receptor and FOXA2 immunostaining).
- Figure 4A-D results are similar to those reported in Figure 5B-C (endometrium) and Figure F-G (lungs)
- Figure 4E-F results are similar to those reported in Figure I-L
- Kaplan-Meyer in Figure 4O is similar to that shown in Figure 5M.
- Figure 5A-B, Figure 5E-F are similar to Figure 6A-B.

We thank the reviewer for their constructive comments regarding our manuscript. We recognize that our original submission failed to highlight the significant advancements in our study beyond those presented by Kriseman et al., 2019. We carefully considered this observation and have restructured our figures to emphasize the new mechanistic data generated in endometrial organoids as part of these studies. In the revised manuscript, Figures 4, 5 and 6 have been reorganized and now present key data that had been previously included in the Supplemental Files. Furthermore, Figures 1, 3 and 4 from our original submission are now in the Supplemental Figures. We hope that the new presentation of our data highlights not only the significance of our studies but also emphasizes the crucial roles of SMAD2/3 in the control of endometrial regeneration/differentiation and endometrial homeostasis.

2. We also found some apparent discrepancies between the results presented in Kriseman et al., 2019 and the present study (may be caused by different Cre mice used) :
Present study :

“....Similar to the 12-week timepoint, there was no observed change in FOXA2 immunoreactivity between *Smad2/3* cKO and control mice at 6 months of age (Figure 3Q-T).”

Kriseman et al., 2019:

« Compared with controls (Fig. 4I and J), disordered uterine glandular structures in the *Smad2/3*cKO mice were visualized by immunohistochemistry of the glandular marker FOXA2 (Fig. 4K and L), with a corresponding increase in *Foxa2* gene expression in the *Smad2/3*cKO mice relative to the controls (Fig. 4D) »

We appreciate this important observation and agree that there are differences in the results for FOXA2 between the two lines (*Smad2/3*-PRcre from Kriseman et al. and *Smad2/3*-LTFcre from this manuscript). This raises important differences between the two mouse lines due to the expression of the cre and timing of the onset of cre-recombinase activation. Specifically, while PRcre is active in the endometrial stroma and epithelium, LTFcre is active only in the endometrial epithelium. Another key difference relates to the timing of cre-recombinase activation, where PRcre is active earlier during postnatal endometrial development than LTFcre. These points are now addressed in the Discussion section:

Lines 488-501: “We previously generated mice with conditional SMAD2/3 deletion using PR-cre (“SMAD2/3-PRcre”) and also observed development of metastatic endometrial cancer¹⁸. However, the onset of hyperplasia and tumor development was more rapid in the SMAD2/3-PRcre mice than in the SMAD2/3-Ltf-cre model presented here. We also observed differences in expression of the glandular epithelial marker, FOXA2⁶⁶, where the SMAD2/3-PRcre mouse line expressed higher levels of FOXA2 than age-matched controls but the SMAD2/3-Ltf-cre mice did not. This discrepancy in FOXA2 expression between the two mouse lines can be explained by the fact that the SMAD2/3-PRcre mice developed endometrial hyperplasia much sooner (by 6 weeks of age) than the SMAD2/3-Ltf-cre mice (by 12 weeks of age), causing expression of FOXA2, a glandular marker, to appear more prominent than controls in the SMAD2/3-PRcre model. Furthermore, differences in the cell types expressing cre (stromal/epithelial in PR-cre vs. epithelial-specific in Ltf-cre), or the earlier onset of PRcre recombinase activity compared to Ltf-cre, could also account for the differences noted in FOXA2 expression.”

Other comments:

3. Figure 2E-G: an immunohistochemistry with p-SMAD3 or p-SMAD2/3 should be performed.

We appreciate the importance of this comment. To address it, we have performed immunostaining of pSMAD2/3 with an antibody that recognizes the phosphorylated epitopes of both SMAD2 and SMAD3. These figures are now displayed in Figure 1E-H.

4. Regarding the second part of the manuscript, Figure 8 shows an scheme of the putative roles of TGF in the endometrium, leaving only Figures 6 and 7 as the only ones showing experimental data experimental. In Figure 6, the authors show an increase of retinol metabolism and downregulation of Wnt signaling identified by RNA-seq experiments. Although the results are potentially interesting, there is no functional assessment of the role these pathways on endometrial organoids.

We thank the reviewer for this helpful observation. In this revision, we have modified the organization of our figures to highlight the novel data that was generated in this manuscript and to demonstrate the methodological and conceptual differences from our previous publication by Kriseman et al., (PMID 30651315). Our major changes in the Figure presentation include the following:

- 1) The original Figure 1 was moved to a new Supplemental Figure 1.
- 2) The original Figures 3 and 4 were moved to new Supplemental Figures 2 and 3.
- 3) The original Figure 6 and portions of the original Supplemental Figures 3, 4 and 5, were restructured to create three new main Figures in the text, Figures 4, 5 and 6.

We anticipate that these extensive changes and the newly organized figures highlight our new work with endometrial organoids. Specifically, we believe that these changes display the morphological differences found due to SMAD2/3 inactivation, and emphasize our findings related to the differential gene expression pathways and genome wide SMAD4 binding in endometrial organoids with impaired SMAD2/3 signaling.

Minor points:

5. Figure 1 is a c-bioportal search showing no experimental-based data. This figure should be moved to a supplementary material.

Thank you for the helpful suggestion. We have moved this figure to the supplementary material.

6. Figures 5E-H are not cited in the text.

Thank you for pointing this out. These figures (revised Figure 3E-H), are now cited in the text, as follows:

Lines 235-239: *“Cross-sections of the uteri revealed that the untreated ovariectomized control, Smad2/3 cKO mice, and the ovariectomized controls treated with E2, retained a properly organized endometrial compartment (Figure 3E-G). However, ovariectomized Smad2/3 cKO mice treated with E2 displayed expansion of the epithelial cells in the tumor mass (Figure 3H).”*

REVIEWERS' COMMENTS:

Reviewer #2 (Remarks to the Author):

The authors addressed all may main concerns.